# Virtual Care Appointments and Experience Among Older Rural Patients with Chronic Conditions in New South Wales: An Analysis of Existing Survey Data

**DOI:** 10.3390/ijerph21121678

**Published:** 2024-12-17

**Authors:** Eloise A. B. Price, Mohammad Hamiduzzaman, Vanette McLennan, Christopher Williams, Victoria Flood

**Affiliations:** 1Faculty of Medicine and Health, The University of Sydney, Sydney, NSW 2006, Australia; 2University Centre for Rural Health, Faculty of Medicine and Health, The University of Sydney, Lismore, NSW 2480, Australiavicki.flood@sydney.edu.au (V.F.)

**Keywords:** virtual care, telehealth, older adults, chronic conditions, rural health, Australia

## Abstract

This retrospective, descriptive study, conducted in 2024, analysed Virtual Care Survey (2020–2022) data of patients’ self-reported reflections on use and experiences to investigate relationships between demographics, the number of chronic conditions, and virtual care use among older rural patients (≥65 years with at least one chronic condition) living in New South Wales, and their satisfaction with virtual care. Associations between categorical variables were assessed using chi-squared tests, and Kruskal–Wallis tests were used for continuous variables. Qualitative feedback was analysed thematically. The study included 264 patients (median age 74 years; 51.1% women). Most virtual care appointments (65.3%) were for consultations, check-ups, or review of test results. Over one-third (38.3%) of the patients had multimorbidity and were 1.8 times more likely to have five or more virtual care appointments compared to the patients with one chronic condition. The oldest age group (≥80 years) preferred telephone over online mediums (Skype or Zoom) (*p* < 0.05). Patient satisfaction was high (65.8%), with 60.9% finding virtual care comparable to in-person consultations. Technological issues correlated with more negative experiences (*p* < 0.05). Key themes were enhanced accessibility and convenience, quality and safety of virtual care, and recommendations for equitable access. Despite positive responses, addressing technological complexities is important for optimising virtual care models for older rural Australians with chronic conditions.

## 1. Introduction

Chronic conditions are common in old age, affecting both the physical health and the nervous system, disrupting daily activities, and diminishing overall quality of life. They last for at least six months, differ in aetiology and symptoms, and represent a wide range of conditions such as cancer, heart disease, stroke, arthritis, lung disease, chronic kidney disease, type 2 diabetes, and neurological and mental disorders [1]. Collectively, these conditions account for 74% of global deaths [2]. In Australia, the prevalence of chronic conditions is high, with 80% of older adults (≥65 years) reported to have at least one chronic condition and over a quarter having three or more, resulting in an economic burden of $38 billion annually [3,4]. In response to the increasing burden of chronic conditions on the overall health system, the Australian Government adopted a National Strategic Framework for Chronic Conditions in 2019, providing policy directions for the prevention and management of these conditions. Since the COVID-19 pandemic, a range of virtual care services has been introduced to enhance access to healthcare services, particularly for disadvantaged populations. To ensure the delivery of safe and high-quality virtual care, the Australian Government introduced the National Digital Health Strategy 2023–2028, creating a more connected and digitally enabled healthcare system [5].

Virtual care plays an integral role in delivering healthcare services to older patients with chronic conditions [6]. Virtual care appointments describe the remote interaction between healthcare professionals and patients, facilitating the delivery of healthcare services, information, and resources through digital technologies, including platforms such as telemedicine, remote patient monitoring, portal messaging, electronic prescriptions, and other innovations like mobile health apps, chatbots for health advice, and virtual health coaching [6]. In addition, it allows real-time videoconferencing between patients and their healthcare providers [6]. Clinical benefits of using virtual platforms are well documented, showing improvements in heart failure and respiratory symptoms, diabetes control, managing blood pressure in hypertensive patients, and reduced hospitalisations [7,8,9]. Beyond clinical outcomes, virtual care offers non-clinical benefits, such as increased attendance in appointments, improved health knowledge and self-management of chronic conditions, and a sense of adequate support [10]. Given the advantages, the use of virtual care is expected to grow across Australian states and territories, especially in rural towns where a shortage of health staff is evident [11].

Approximately 34% of older Australians live in rural towns, with a higher prevalence of chronic conditions than their metropolitan counterparts [12,13]. Older rural Australians with chronic conditions often require ongoing management of their conditions, regular check-ups, management of medication, and information about available services [14]. While virtual care presents a solution to meet healthcare needs by reducing the necessity to travel, there are challenges for both patients and providers [15]. These challenges include limitations in physical examination, increased workload on health staff, reification of patient-healthcare provider relationships, restricted non-verbal communication, and privacy concerns [16,17,18,19]. Rural patients face additional challenges such as a lack of understanding of their complex healthcare needs among tele-healthcare providers, limited access to the latest technologies, and issues with internet availability and connections [19]. The New South Wales (NSW) Government is pioneering the delivery of healthcare services to rural and remote patients through virtual care platforms by implementing its Virtual Care Strategy 2021–26 to address the challenges in virtual care [20].

While available data highlights older Australians’ increasing use of the internet for various purposes, such as email (95%), banking (77%), and shopping (64%), evidence on the digitalisation trends specific to virtual care in rural NSW is limited and fragmented [21]. Disparities in digital literacy persist, particularly in rural areas. For example, about 34% of Australians aged 50 and above demonstrate low digital literacy, and a higher proportion of them are from rural communities [11]. Virtual care is one of the many digital interactions with public services, demonstrating benefits for patients in NSW [6,21]. This underscores the importance of further evaluation of its usage, quality, and safety for managing chronic conditions among older rural patients.

The Bureau of Health Information (BHI) has conducted a virtual care survey annually in NSW since 2020. This study uses existing BHI survey data on patients’ use and experiences of virtual outpatient care appointments to investigate the following:-The relationship between demographics, the number of chronic conditions, and virtual care use among older rural patients (≥65 years with at least one chronic condition) living in NSW.-Experience and satisfaction with virtual outpatient care among rural older patients with chronic conditions in NSW.

## 2. Materials and Methods

### 2.1. Study Design

This is a retrospective, descriptive study conducted in 2024, analysing a Virtual Care Survey dataset of patients’ self-reported reflections on the use and experiences of their recent virtual outpatient care. The original survey captures patients’ demographic data (i.e., postcode, age, gender, education, language, and chronic health conditions); most recent virtual care appointments (i.e., self-reported purpose of appointment, types of health services, virtual care platform, connection problem/s, technical support); and reflections of experience and satisfaction with care over the past 12 months (i.e., confidence and trust, number of appointments, benefits of virtual care, challenges of virtual care, and best part of virtual care experiences). The survey also included two free-text questions: “what was the best part of your virtual care experience?” and “what most needs improving about your virtual care experience?”.

### 2.2. Data Sources

We used the de-identified unit record data relating to virtual care from the BHI’s NSW Patient Survey Program. Data from three Virtual Care Surveys conducted between 2020 and 2022 were combined for this study. In the BHI’s NSW Patient Survey Program, the data collection process involved stratified random sampling, with invitation letters sent to selected participants, followed by reminder letters two weeks later to maximise response rates and ensure robust sampling, as detailed in the Virtual Care Survey technical supplement [22]. Survey responses were anonymised, underwent quality assurance checks, and were securely stored on password-protected servers accessible only to authorised staff.

Following a Data Sharing Agreement between BHI and the corresponding author (MH), data were accessed within the Secure Unified Research Environment (SURE) platform [23]. Both the first author (EP) and corresponding author (MH) completed the SURE Researcher Training Program (SURE-UT-001), which covered setting up a virtual workstation for SURE access and managing data within the platform. Separate usernames and passwords were provided by the SURE admin to EP and MH for accessing the virtual care dataset.

### 2.3. Defining the Study Cohort

In total, 7735 patients completed the survey across the three years. To answer the research questions of this study, we focused on a specific cohort that met the following inclusion criteria:-patients aged 65 years or older,-living in a rural town,-having at least one chronic condition (lasting for at least six months), and-had at least one virtual care visit with a health practitioner in the past 12 months.

Exclusion criteria included patients with no virtual care appointments in the past 12 months or those unable to provide consent or complete the survey due to language or cognitive barriers.

A low number of patients were eligible to be included in the study because of the application of specific inclusion criteria, particularly the focus on rural older patients with chronic conditions who had at least one virtual care appointment completed. While the original dataset included all virtual care patients across NSW, a large proportion were from major cities and thus excluded from this analysis. The smaller population of rural older patients eligible for virtual care services led to the reduced sample size.

### 2.4. Variables

Self-reported chronic conditions were pre-defined by the following: longstanding (>6 months) chronic illness (e.g., cancer, diabetes, chronic heart disease), longstanding physical condition (e.g., arthritis, spinal injury, multiple sclerosis), intellectual disability, mental health condition (e.g., depression, anxiety), neurological condition (e.g., Alzheimer’s, Parkinson’s disease). Chronic conditions were then dichotomised into a variable for the number of chronic conditions (“one chronic condition” and “multimorbidity as more than one chronic health condition”).

The patients were defined as being in rural areas according to their residential postcode. Rurality was defined according to the Australian Statistical Geography Standard, based on access to services measured using the Accessibility/Remoteness Index of Australia Plus (ARIA+) [24]. Only patients who were “outer-regional, remote, or very remote” were included.

Differences in patient characteristics were assessed based on age, gender, level of education, and socio-economic status. The Index of Relative Socio-Economic Disadvantage was used as an indicator of relative socioeconomic advantage and disadvantage [25]. This was classified into quintiles, with the lowest quintiles indicating the highest level of disadvantage.

Outcome variables used to define patients’ use of virtual care included the self-reported purpose of virtual care appointments over the past 12 months, the number of virtual care appointments, the healthcare professional seen, and the type of virtual care appointment. Patients’ satisfaction and experience with virtual care were reported in the survey in response to free-text questions. It was possible to select more than one answer on the survey for the purposes of virtual care appointments and type of chronic condition.

### 2.5. Data Analysis

Data were analysed on the SURE platform. All quantitative analyses were conducted using IBM SPSS Statistics (Version 29). The demographics of respondents were described in terms of age, gender, level of education, and socio-economic disadvantage. The demographic data and number of chronic conditions were then compared against virtual care usage variables. Categorical data were presented as frequencies and proportions, with normally distributed data summarised by mean and standard deviations and non-normally distributed data with median and interquartile ranges. The Pearson chi-squared (χ2) test was used to assess the association between categorical variables. Due to the age distribution being skewed, a non-parametric Kruskal–Wallis test compared age between variables with three or more groups. A *p*-value of <0.05 was considered statistically significant; however, the interpretation also reflected the clinical relevance of observed differences and the power of the study.

Satisfaction and experience with virtual care were evaluated, and a qualitative analysis of free-text survey responses was conducted using reflexive thematic analysis, following Braun and Clarke’s (2021) six-step approach for identifying and analysing patterns within qualitative data [26]. This approach starts with familiarisation, where the first author (EP) repeatedly reads the data and makes notes. Next, key segments are labelled with codes and discussed with MH and VM in regular research team meetings before grouping them into categories. The categories were then reviewed by the broader team for the identification of sub-themes and themes. Finally, the themes were refined to ensure accuracy, defined and named to clarify their meaning, and reported in detail, supported by examples, and linked to the research question.

### 2.6. Ethics Approval

Approval to conduct this study was obtained by the University of Sydney Human Research Ethics Committee on 29 April 2024 (Approval number: 2024/HE000207).

## 3. Results

### 3.1. Quantitative Findings

#### 3.1.1. Demographic Characteristics of Participants

Here, 264 older rural patients were eligible for inclusion in the analysis (Figure 1). The median age of participants was 74.0 years (IQR: 70.0–78.0), with a significantly higher median age in 2021 (75.5 [IQR: 70.0–81.0]) compared to 2020 (74.0 [IQR: 70.0–78.0]) and 2022 (73.0 [IQR: 69.0–77.0]); *p* = 0.026. In 2021, 15.3% (19/124) of patients were aged 65 to <70 years, compared to 30.2% (19/63) in 2020 and 29.9% (23/77) in 2022. Overall, 51.1% (135/264) of patients were women, with no significant difference across years.

About 64.0% (169/264) of the patients had a longstanding chronic illness, while 54.5% (144/264) had a longstanding physical condition, 16.3% (43/264) had a mental health condition, and 10.2% (27/264) had a neurological condition. Of the 101 patients with multiple chronic conditions, 82.7% (n = 83) had two comorbidities, and 17.8% (n = 18) had three. Self-reported education level showed that 53.6% (141/263) had not completed Year 12, while the proportion of university graduates was higher in 2022 (19.5%, 15/77) compared to 2020 (9.5%, 6/63) and 2021 (9.8%, 12/121); *p* = 0.021. A large proportion of patients lived in areas of low socioeconomic status, with 37.9% (99/261) residing in the most disadvantaged areas (quintile 1), with no significant differences in socioeconomic status across survey years.

#### 3.1.2. Use of Virtual Care

Overall, 241 (91.6%) older rural patients responded to the question about their purpose for attending a virtual care appointment. Patients were able to select more than one option on the survey, creating a total of 382 responses for attending a virtual care appointment. In total, 84 (21.9%) appointments were for an initial or follow-up consultation, 114 (34.7%) were for medical diagnosis or treatment, 79 (24.1%) were for review of treatment or results, and 66 (17.3%) were for a regular check-up (see Table 1). Of the 240 patients who responded to the question on the number of virtual care appointments in the last 12 months, 53.8% (n = 129) had 1 to 2 appointments, 32.5% (n = 78) had 3 to 5 appointments, and 13.8% (n = 33) had more than 5 appointments.

The number and type of virtual healthcare appointments are presented in Table 2. Of the 263 patients responding to the question about the healthcare professional that they saw at their virtual care appointment, 152 (57.8%) consulted a doctor only, and 58 (22.0%) consulted an allied health professional. 20.2% (53/263) of patients saw a multidisciplinary team (more than one type of healthcare professional), and 69.8% (37/53) of these patients consulted both a doctor and a nurse. The most frequent type of virtual care appointment was conducted by telephone (e.g., mobile or landline), which was used by 47.4% (120/253) of patients. Online appointments (e.g., using Skype or Zoom on any device) were used by 35.6% (90/253). There was no significant difference between the number of appointments, healthcare professionals seen, or type of appointment between years.

#### 3.1.3. Relationship Between Demographics of Patients and Their Virtual Care Use

The median age of those who had 1–2 appointments was 74.0 years [IQR: 70.0–80.5] and similar to that of patients who had 5+ appointments (73.0 years [IQR: 69.0–79.0]). There was no difference in the number of appointments made between men and women. Overall, 18.4% (16/87) of patients with more than one chronic condition had five or more appointments compared to 11.1% (17/153) of patients with only one chronic condition (Odds Ratio (OR) = 1.80 [95%CI 0.86–3.78], *p* = 0.168). There was also a trend for patients with a higher education level (University degree) to have a greater number of multiple appointments (5+), but this did not reach statistical significance; see Table 3. There was no clear trend in the number of appointments and socioeconomic status. The relationship between the demographic data of patients and their virtual care use is presented in Table 3.

Patients seeing a multidisciplinary team had a median age of 75.0 [IQR: 70.0–81.5], compared to those seeing only an allied health professional, who had a median age of 72.0 [IQR: 67.0–81.0]; *p* = 0.227. Almost a quarter (23.8% (24/101)) of patients with more than one chronic condition saw a multidisciplinary team compared to 17.0% (29/162) of patients with only one chronic condition; *p* = 0.350. Overall, there were no significant differences between the demographic data and the type of healthcare professional seen at appointment.

Older patients were more likely to use the telephone over online mediums for their appointment (55.1% (38/69) of patients ≥80 years) compared to 29.0% (20/69) who used online; *p ≤* 0.05. Overall, there were no statistically significant differences or trends in the type of virtual care appointment and baseline demographic (Table 3).

#### 3.1.4. Satisfaction with Virtual Care

Overall, 65.8% (171/260) of patients rated their virtual care experience as ‘very good’ and 24.6% (64/260) as ‘good’. Although 6.6% (17/259) reported that the care and treatment they received did not help them, 73.7% (191/259) stated it definitely helped, and 19.7% (51/259) reported it helped to some extent. Comparing virtual to face-to-face consultations, 60.9% (156/256) thought it was about the same, 11.7% (30/256) thought it was better, and 27.3% (70/256) thought it was worse. Overall, 86.8% (223/257) of patients stated they would use virtual care again.

Those who experienced problems with connectivity or technology were more likely to have a negative overall experience with virtual care (*p* < 0.05). Of the patients who experienced problems with connectivity or technology, 56.5% (13/23) rated their experience as ‘very good’ or ‘good’ compared to 94.4% (219/232) of patients who did not experience problems with technology. Of the patients who experienced technological issues, 17.4% (4/23) rated their experience as ‘poor’ or ‘very poor’ compared to 2.2% (5/232) of patients who did not experience technical problems (see Figure 2). Additionally, patients who found their virtual care convenient were more likely to rate their experience positively [96.4% (162/168) vs. 81.9% (68/83); *p* < 0.05].

### 3.2. Qualitative Findings

Three major themes were identified in the free-text responses: enhanced accessibility and convenience, quality and safety of care, and recommended improvements for equitable access. Example quotations are provided in the Table of Themes (see Table 4).

#### 3.2.1. Enhanced Accessibility and Convenience

Most older rural patients with chronic conditions had positive experiences with their virtual care visits, describing it as a “very personal and rewarding experience”, “excellent”, “invaluable” or “satisfying”. These positive comments were attributed to the increased access and convenience that virtual care platforms provide in accessing health services. Virtual appointments helped overcome geographical barriers and provide flexibility that traditional appointments could not. Patients appreciated being able to access “specialist treatments [that are many] …” and were able to “communicate from just about anywhere”. The convenience of not having to travel was beneficial to them, minimising barriers such as distance, time, and cost of travel. For example, the experience of travelling long hours—“[many] hours one way to Sydney for a brief check-up means a whole day of exhausting travel”—could be avoided with virtual care. Some patients appreciated the convenience when too sick to travel, in pain, or isolated at home. For older adults living rurally, mobility issues were an added barrier, and virtual care eliminated some of these, such as the need to “get a walker in and out of the car at both ends of the appointment”.

Reduced stress from not worrying about travel logistics and costs also contributed to a more relaxed healthcare experience. Virtual care appointments saved them the cost of flights and accommodation; for example: “[virtual care appointments] save many hours of travel each way plus overnight accommodation and meals”, especially as they “cannot afford to drive this distance any more”. Patients acknowledged that while in-person appointments are ideal, virtual care is a valuable addition to the health system, especially during the COVID-19 pandemic and was seen as “better than nothing”.

#### 3.2.2. Quality and Safety

Despite the virtual format, many older rural patients felt they received high-quality and safe care, praising the healthcare professionals for being “caring, friendly, and professional” and “excellent at their tasks”. The quality and safety of virtual care as described by patients were related to timely access, empowerment, and technological complexities. Virtual care facilitated timely access to quality care; for example: “[it] allowed [them] to receive advice sooner than having to wait for an appointment”. In rural areas, where hospital wait times can be long, virtual care can be critical. For example: “[it] was critical in saving my life”. In addition, virtual care platforms empowered many patients to self-manage their conditions by providing education and “information on lifestyle requirements to aid management of [their] conditions”, as well as keeping patients up to date on their conditions.

A small group expressed dissatisfaction, preferring face-to-face interactions and feeling that the virtual format did not meet their expectations. For most, their dissatisfaction stemmed from technological issues or that the specialist “missed or was late to appointments”. Many patients struggled with navigating technology, often relying on external support. For some, this support was not always available when accessing care from home. Therefore, the lack of technological access and literacy were significant barriers. However, some thought that the technological issues hindered the quality of care received, and several patients reported that virtual care was not as beneficial as in-person appointments.

#### 3.2.3. Recommendations to Achieve Equitable Access

Older rural patients identified areas for improvement, including providing technical support for elderly patients to help with managing technological failures. Some patients expressed a need for more available appointments and assistance in accessing virtual care in rural areas. Effective communication and interaction were identified as areas needing improvement. Patients wanted to see who they were speaking to, having specialists call on time, better communication from staff, and specialists showing more interest in their treatment questions. The need for more information about post-operative care, more time with their healthcare providers, periodic physical examinations, and house calls for elderly and incapacitated individuals were also emphasised to ensure continuity of care.

## 4. Discussion

At a time when virtual care has become a cornerstone of NSW Health, this analysis of existing survey data provides new insights into older rural patients with chronic conditions in NSW and their self-reported use and experiences of outpatient virtual care encounters and extends previous findings on satisfaction with virtual care. Data were collected during the COVID-19 pandemic, and as expected, older rural patients with chronic conditions often welcomed the use of technology for healthcare access. Lockdowns in 2020 and 2021, though community prevalence was relatively low in rural NSW compared to other countries, were followed by significant pandemic-related activity in 2022 after the reopening of international borders. This may have an impact on their use and experiences with virtual care, as many of them had no choice but to rely on it during that time. While its usage did not significantly vary by age or gender in this cohort, a notable trend emerged in the dataset that is consistent with a national Australian study that found older rural patients with a higher level of education (university degree or higher) were more likely to access virtual care [27]. Similar to current evidence, this study found that older rural patients with chronic conditions reported a high level of satisfaction with virtual care [28]. For rural patients, healthcare access is often complicated by the necessity to travel long distances to regional health centres, which introduces hidden costs, such as fuel expenses, accommodation, and lost income [29]. These factors exacerbate the financial burden of healthcare access in rural areas, making virtual care an attractive way to complement healthcare. Since the COVID-19 pandemic, there has been a global cost-of-living crisis, further intensifying the need for affordable healthcare options like virtual care [30]. In addition, the time savings of virtual care are particularly important for rural older patients with chronic conditions who require regular appointments for script renewals and check-ups [31]. The potential of virtual care is tempered by the realities of its inherent limitations and the rural-specific challenges. Four intriguing research findings of this study are discussed in this section.

Firstly, this study identified that around one-third of the virtual care appointments among older rural patients with chronic conditions were for medical diagnosis and treatment. Virtual care in Australia saw widespread use of telehealth and video consultations, but its primary applications are focused on initial consultations, follow-ups, and ongoing management of chronic conditions rather than medical diagnosis. For instance, Isautier et al. (2020) found that patients often used telehealth for follow-ups and medication management [32]. Similarly, the Medicare Benefits Schedule (MBS) Activity Report showed that a significant proportion of telehealth consultations were for ongoing care rather than diagnostic purposes [33]. This is likely because of the fact that many diagnostic assessments, especially for those with chronic conditions, require physical examinations, and one inherent limitation of virtual care is its inability to perform such examinations [16]. An in-depth investigation of this issue is beyond the scope of this descriptive study, but it is important that diagnosis and treatment issues are addressed in virtual care models. Remote monitoring is occurring through wearable devices that can track vital signs, blood sugar levels, and other physiological metrics, critical to the early detection of hypertension and diabetes [34]. There are expectations for integrating a blend of face-to-face and virtual care approaches as suggested by the NSW Virtual Care Strategy or for incorporating wearable devices or ambulatory clinics into virtual care programs to support sustained virtual care for chronic conditions rurally [20].

Secondly, this study identified that older rural patients with chronic conditions, especially those in the older age group (≥80 years), preferred telephone appointments over video consultations (e.g., Skype or Zoom). Previous studies have identified connectivity issues and confidence in technology as significant barriers to the use of video consultations [35,36]. A systematic review on the effectiveness of telephone versus video consultations reported video consultations outperformed telephone consultations in 50% of the published studies, reporting superior clinical, service, or economic outcomes. While many simple consultations can be effectively managed via telephone, video consultations are beneficial for patients with complex and chronic care needs where video examination may be required [37]. However, while video consultations provide a more comprehensive visual medium, allowing for improved examination, patient interaction, and care, these benefits are only achievable when patients have the necessary digital literacy and stable access to technology [38]. In rural areas with poor connectivity or limited digital confidence, patients may be reluctant or unable to engage with video consultations, defaulting to telephone appointments as the more accessible and practical option. Therefore, despite the clinical advantages of video consultations, addressing these technological barriers remains essential for improving adoption among older, rural populations.

The third major finding is that virtual care was used more frequently in those with multimorbidity, and although this was not statistically significant, likely due to small numbers, there was a trend toward more frequent use for those patients. Previous studies such as Singer et al. (2022) and Stamenova et al. (2022) confirmed the higher utilisation of virtual care platforms among patients with chronic conditions, particularly those with multimorbidity [39,40]. Patients with multimorbidity have increased healthcare needs and require a well-coordinated system of care across multiple providers in different settings [41]. Reflected in this study was a trend of patients with multimorbidity seeing a multidisciplinary team compared to those with one chronic condition. In recent years, virtual care has become increasingly recognised as a proactive healthcare strategy to integrate and improve care for the complex needs of multimorbid patients [42]. A subtheme identified in this study was that patients believed virtual care allowed them to self-manage their conditions. Virtual care facilitates self-management of chronic conditions by providing regular monitoring, personalised education, medication management, lifestyle and behavioural support, enhanced communication, and increased accessibility [43]. Together, these strategies empower patients to take control of their health, leading to better outcomes and an improved quality of life [44]. This is consistent with previous studies that have identified non-clinical benefits of virtual care use in patients with chronic conditions, such as improved appointment attendance, increased patient knowledge and self-management, and patients feeling more supported [45,46].

There is a need to address technological complexities and connectivity issues to optimise the virtual care models for older rural adults with chronic conditions [47,48]. These barriers may be magnified in patients residing in lower socioeconomic areas [37] or patients who struggle to use technology [47]. When comparing the demographic data of the study population with their use of virtual care, there was a trend for older patients having a lower use of online appointments. This highlights the impact of the ‘digital divide’, which includes discrepancies in internet accessibility, computer availability, and digital literacy [38]. Older patients may not prioritise technology acquisition, and even if they have access to technology, they may lack the knowledge to use it effectively for video consultations or other telehealth modalities [48,49]. This study found that patients who experienced technical difficulties were more likely to have an overall negative experience of virtual care. Technical issues have been previously recognised in many studies as a barrier to virtual care satisfaction [35,50,51].

A key strength of this study was its analysis of the perspectives of older rural patients with chronic conditions on virtual care, including qualitative insights into its limitations. However, this study has several limitations. Rurality was broadly categorised into major cities, inner regional, and a combined category for outer regional, remote, and very remote areas, limiting the ability to differentiate between these diverse rural experiences. The small sample size (n = 264) further posed challenges for statistical comparisons, potentially reflecting cognitive bias across the study’s three-year period. The survey was conducted in English, which may have contributed to the exclusion of patients from culturally and linguistically diverse communities. Additionally, due to the cross-sectional nature of the survey, it is possible some patients completed the survey in consecutive years, introducing bias and inflating the sample size. Additional targeted surveys with larger samples targeting older rural patients with chronic conditions would be of benefit to inform insights on this topic.

## 5. Conclusions

This retrospective study demonstrates that virtual care holds potential in the management of chronic conditions among older rural patients in NSW, focusing on demographics and specific care needs of rural older patients. By enhancing accessibility and facilitating timely access to healthcare services, virtual care can substantially reduce the burdens on older rural patients, their family carers, health staff, and the overall healthcare system. Effective implementation, however, of virtual care in rural towns requires supportive health policies that ensure these programs are contextualised to address technological issues. Further studies should focus on long-term outcomes and the continuous improvement of virtual care interventions, considering the inherent limitations of this approach for managing chronic conditions. Clinically, virtual care providers must be well-trained in both technical and patient engagement aspects to deliver effective, personalised care to this demographic.

## Figures and Tables

**Figure 1 ijerph-21-01678-f001:**
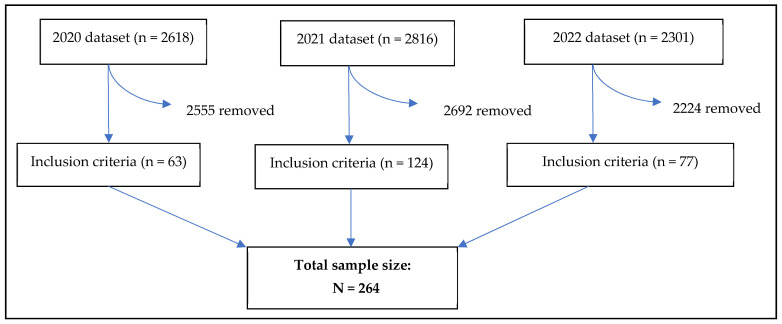
Linkage of data sources—The dataset was provided by BHI as three separate datasets for each year (2020, 2021, 2022). These datasets included all individuals who completed the survey across the three years. Patients were removed from the dataset if they did not meet the inclusion criteria; aged 65 years and over, had at least one chronic condition, and lived in a rural setting. This resulted in sample sizes of n = 63, n = 124, and n = 77 across the years, respectively. The three datasets were then appended, creating a total sample size of 264.

**Figure 2 ijerph-21-01678-f002:**
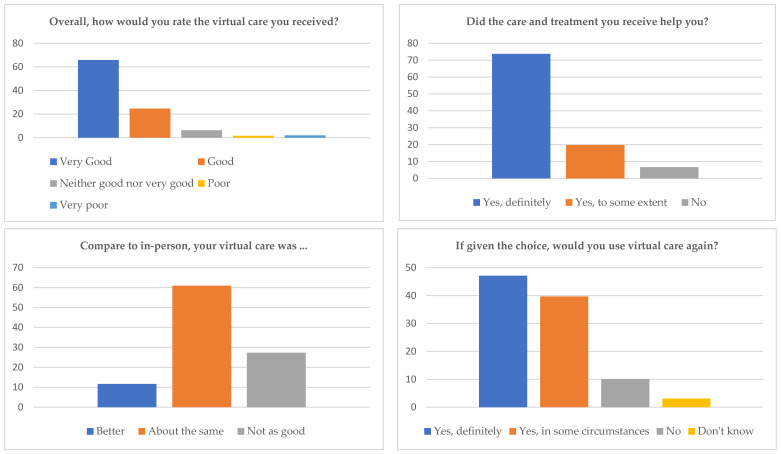
Percentage of patients’ satisfaction and experiences with virtual care.

**Table 1 ijerph-21-01678-t001:** Self-reported purpose of most recent appointment with a hospital outpatient clinic.

Self-Reported Purpose of Virtual Care Appointment	2020		2021		2022		Overall	
Initial/follow up consultation	26	25.7%	33	19.4%	25	22.3%	84	21.9%
Medical diagnosis and treatment (or therapy)	26	25.7%	50	29.4%	38	33.9%	114	34.7%
Review of treatment or results	32	30.5%	29	17.1%	18	16.1%	79	24.1%
Regular check up	14	13.3%	34	20.0%	18	16.1%	66	17.3%
Other	6	5.7%	24	14.1%	13	11.6%	43	13.1%
Total	104	100%	170	100%	112	100%	386 *	100%

* Patients were able to select more than one purpose for their appointment. Of the 261 people who answered this question, 188 people selected one purpose, 40 selected two, and 20 selected three purposes.

**Table 2 ijerph-21-01678-t002:** Virtual healthcare use in older adults from 2020 to 2022.

	2020		2021		2022		Overall		Statistical Test	*p*-Value
Number of virtual care appointments in last 12 months ^a^										
1 to 2	25	41.7%	68	61.8%	36	51.4%	129	53.8%	χ^2^_(df=4)_	0.114
3 to 5	26	43.3%	30	27.3%	22	31.4%	78	32.5%	7.45	
More than 5	9	15.0%	12	10.9%	12	17.1%	33	13.8%		
Healthcare professional seen at virtual care appointment ^b^										
Doctor/specialist	32	50.8%	75	61.0%	45	58.4%	152	57.8%	χ^2^_(df=6)_	0.481
Allied health professional ^c^	20	34.5%	22	37.9%	16	27.6%	58	22.0%	5.50	
Multidisciplinary team ^d^	11	17.5%	26	21.1%	16	20.8%	53	20.2%		
Type of virtual care appointment ^e^										
Telephone ^f^	27	44.3%	60	52.2%	33	42.9%	120	47.4%	χ^2^_(df=4)_	0.728
Online ^g^	23	37.7%	38	33.0%	29	37.7%	90	35.6%	2.04	
Other	11	18.0%	17	14.8%	15	19.5%	43	17.0%		

^a^ Overall, there were 24 missing values for the Number of virtual care appointments in the last 12 months. This made the sample size for each consecutive year n = 60, n = 110 and n = 70, respectively. These missing individuals were not included in the summary statistics for this variable or further analysis involving this variable. ^b^ Overall, there was 1 missing value for the variable Healthcare professional seen at virtual care appointments. This made the sample size for 2021 n = 123. This missing individual was not included in the summary statistics for this variable or further analysis involving this variable. ^c^ Defined as seeing one type of allied health professional only. This includes nurses, physiotherapists, speech pathologists, occupational therapists, social workers, mental health professionals, radiographers, dieticians and podiatrists. ^d^ defined as seeing more than one type of healthcare professional across disciplines. ^e^ Overall, there were 11 missing values for the variable Type of virtual care appointment. This made the sample size for each consecutive year n = 61, n = 115 and n = 77, respectively. These missing individuals were not included in the summary statistics for this variable or further analysis involving this variable. ^f^ Telephone was defined as using a landline or mobile using audio only. ^g^ Online was defined as using an online platform such as Skype, Zoom or a virtual care portal on any device such as a computer, tablet or smartphone.

**Table 3 ijerph-21-01678-t003:** Associations between demographics and virtual care use over the past 12 months.

**(Number of Virtual Care Appointments)**									
	**1–2**		**3–5**		**5+**		**Overall**	**Statistical Tests**	***p*-Value**
Age									
Median, IQR	74.0, [70.0–80.5]		73.0, [69.0–79.0]		73.0, [69.0–79.0]		75.0, [74.2–75.9]	H_(df=2)_ 1.80	0.406
65 to <70 years	26	47.3 *	21	38.2	8	14.5	55	χ^2^_(df=6)_	0.891
70 to <75 years	39	52.7	24	32.4	11	14.9	74	2.30	
75 to <80 years	26	54.2	16	33.3	6	12.5	48		
≥80 years	38	60.3	17	27.0	8	12.7	63		
Gender (n, %)									
Men	63	52.9	38	31.9	18	15.1	119	χ^2^_(df=2)_	0.828
Women	66	54.5	40	33.1	15	12.4	121	0.377	
Number of chronic conditions (n, %)									
One chronic condition	86	56.2	50	32.7	17	11.1	153	χ^2^_(df=2)_	0.270
Multimorbidity (more than one chronic health condition)	43	49.4	28	32.2	16	18.4	87	2.62	
Highest level of education (n, %)									
Less than year 12 or equivalent	69	55.2	40	32.0	16	12.8	125	χ^2^_(df=6)_	0.276
Completed year 12 or equivalent	8	47.1	5	29.4	<5			7.52	
Trade, diploma or certificate	34	50.7	27	40.3	6	9.0	67		
University degree or higher	18	58.1	6	19.4	7	22.6	31		
Level of socioeconomic disadvantage (n, %) ^a^									
Quintile 1 (Most disadvantaged)	55	61.8	23	25.8	11	12.4	89	χ^2^_(df=6)_	0.356
Quintile 2	48	46.6	40	38.8	15	14.6	103	6.63	
Quintile 3	19	50.0	13	34.2	6	15.8	38		
Quintile 4	6	75.0	<5		<5				
**(Healthcare Professional Seen at Virtual Care Appointment)**									
	**Doctor/Specialist**		**Allied Health ^b^**		**Multidisciplinary Team ^c^**				
Age									
Median, IQR	74.0 [70.0–80.0]		72.0 [67.0–81.0]		75.0 [70.0–81.5]		74.0 [70.0–80.0]	H_(df=3)_ 4.34	0.227
65 to <70 years	2	47.5	22	36.1	10	16.4	61	χ^2^_(df=9)_ 14.1	0.119
70 to <75 years	49	64.5	13	17.1	14	18.4	76		
75 to <80 years	31	58.5	9	17.0	13	24.5	53		
≥80 years	43	58.9	14	19.2	16	21.9	73		
Gender (n, %)									
Men	78	60.9 *	22	17.2	28	21.9	128	χ^2^_(df=3)_	0.252
Women	74	54.8	36	26.7	25	18.5	135	4.09	
Number of chronic conditions (n, %)									
One chronic condition	95	58.6	38	23.5	29	17.9	162	χ^2^_(df=3)_	0.350
Multimorbidity (more than one chronic health condition)	57	56.4	20	19.8	24	23.8	101	3.28	
Highest level of education (n, %)									
Less than year 12 or equivalent	83	58.9	31	22.0	27	19.1	141	χ^2^_(df=9)_	0.953
Completed year 12 or equivalent	12	63.2	<5		<5			3.25	
Trade, diploma or certificate	38	55.1	15	21.7	16	23.2	69		
University degree or higher	19	57.6	9	27.3	5	15.2	33		
Level of socioeconomic disadvantage (n, %) ^a^									
Quintile 1 (Most disadvantaged)	61	62.2	19	19.4	18	18.4	98	χ^2^_(df=9)_	0.218
Quintile 2	62	56.4	23	20.9	25	22.7	110	11.9	
Quintile 3	22	50.0	16	36.4	6	13.6	44		
Quintile 4	5	62.5	<5		<5				
**(Type of Virtual Care Appointments)**									
	**Telephone ^d^**		**Online ^e^**		**Other ^f^**				
Age									
Median, IQR	74.0 [69.0–80.0]		73.0 [70.0–78.3]		76.0 [72.0–80.0]		74.0 [69.0–79.0]	H_(df=1)_ 0.240	0.625
65 to <70 years	33	55.0 *	21	35.0	6	10.0	60	χ^2^_(df=6)_	0.023
70 to <75 years	29	38.7	35	46.7	11	14.7	75	14.7	
75 to <80 years	20	40.8	14	28.6	15	30.6	49		
≥80 years	38	55.1	20	29.0	11	15.9	69		
Gender (n, %)									
Men	56	45.9	46	37.7	20	16.4	122	χ^2^_(df=2)_	0.792
Women	64	48.9	44	33.6	23	17.6	131	0.468	
Number of chronic conditions (n, %)									
One chronic condition	75	47.8	60	38.2	22	14.0	157	χ^2^_(df=2)_	0.224
Multimorbidity (more than one chronic health condition)	45	46.9	30	31.3	21	21.9	96	2.99	
Highest level of education (n, %)									
Less than year 12 or equivalent	64	48.1	40	30.1	29	21.8	133	χ^2^_(df=6)_	0.297
Completed year 12 or equivalent	10	52.6	8	42.1	<5			7.26	
Trade, diploma or certificate	32	47.1	27	39.7	9	13.2	68		
University degree or higher	14	42.4	15	45.5	<5				
Level of socioeconomic disadvantage (n, %) ^a^									
Quintile 1 (Most disadvantaged)	43	46.7	36	39.1	13	14.1	92	χ^2^_(df=6)_	0.465
Quintile 2	52	48.1	34	31.5	22	20.4	108	5.64	
Quintile 3	17	40.5	18	42.9	7	16.7	42		
Quintile 4	6	75.0	<5		<5				

^a^ There were no patients in this study who resided in quintile 5. ^b^ Allied health professional has been defined as seeing one type of allied health professional only. This includes nurses, physiotherapists, speech pathologists, occupational therapists, social workers, mental health professionals, radiographers, dieticians and podiatrists. ^c^ Multidisciplinary team has been defined as seeing more than one type of healthcare professional across disciplines. ^d^ Telephone was defined as using a landline or mobile using audio only. ^e^ Online was defined as using an online platform such as Skype, Zoom or a virtual care portal on any device such as a computer, tablet or smartphone. ^f^ Other was not defined in the survey question. * Percentages are calculated within each row and indicate the proportion of each type of healthcare professional seen within each demographic category.

**Table 4 ijerph-21-01678-t004:** Summary of themes and subthemes from the Qualitative Analysis of Patient Experiences and Satisfaction with Virtual Outpatient Care.

Themes	Subthemes
Enhanced accessibility and convenience	Better access to health services
	Convenience
Quality and safety	Timely access to care
	Clinical self-management
	Technological issues as a barrier to access
	Negative experiences
Recommendations to achieve equitable access	Technical support for elderly patients
	Addressing inequities in access to care for rural areas
	Communication and interaction
	Follow-up and continuity of care
	satisfaction and no improvement needed

## Data Availability

Restrictions apply to the availability of these data. Data were obtained from the Bureau of Health Information and are available with the permission of the Bureau of Health Information.

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
