# Peer review of "Virtual Care Appointments and Experience Among Older Rural Patients with Chronic Conditions in New South Wales: An Analysis of Existing Survey Data"

_ijerph, 2024, doi:10.3390/ijerph21121678_

Round 1
Reviewer 1 Report
Comments and Suggestions for Authors
I liked the article and I think it provides a vision of what virtual care can offer to chronic patients.
On page 120, the low number of patients recruited in three years is striking. In order to assess the representativeness of the sample, we would need to know the target population of this region and its diseases from which this cohort of included patients is drawn. It seems strange that only 3% of the patients included in the program meet the inclusion criteria. I think this should be explained.
Author Response
Comment: I liked the article and I think it provides a vision of what virtual care can offer to chronic patients.
Response: Thank you.
Comment: On page 120, the low number of patients recruited in three years is striking. In order to assess the representativeness of the sample, we would need to know the target population of this region and its diseases from which this cohort of included patients is drawn. It seems strange that only 3% of the patients included in the program meet the inclusion criteria. I think this should be explained.
Response: 7,735 patients completed the original survey across the three years. The low number of patients included in the study reflects the study objectives and the application of specific inclusion criteria, particularly the focus on rural older patients with chronic conditions who had at least one virtual care appointment completed. While the original dataset included all virtual care patients across NSW, a large proportion were from major cities and thus excluded from this analysis. The smaller population of rural older patients eligible for virtual care services led to the reduced sample size for this study.

Reviewer 2 Report
Comments and Suggestions for Authors
1. Abstract - adequate and clear
2. Introduction - The introduction highlights the current situation of telemonitoring in Australia. The intro content is a short and precise presentation.
3. Methods - Explained and presented clearly. However, an explanation of the huge removal of data collection must be justified.
4. Discussion - The discussion was explained and presented well.
5. Conclusion - the article was concluded well.
Author Response
|
Comment: Abstract - adequate and clear.
Response: Thank you.
Comment: Introduction - The introduction highlights the current situation of telemonitoring in Australia. The intro content is a short and precise presentation.
Response: Thank you.
Comment: Methods - Explained and presented clearly. However, an explanation of the huge removal of data collection must be justified.
Response: Thank you. The huge removal of data was because of the study purpose and the application of specific inclusion criteria, particularly the focus on rural older patients with chronic conditions who had at least one virtual care appointment completed. While the original dataset included all virtual care patients across NSW, a large proportion were from major cities and thus excluded from this analysis. The smaller population of rural older patients eligible for virtual care services led to the reduced sample size. [Page 3; Line number: 129-134]
Comment: Discussion - The discussion was explained and presented well.
Response: Thank you.
Comment: Conclusion - the article was concluded well.
Response: Thank you. |

Reviewer 3 Report
Comments and Suggestions for Authors
I would like to congratulate the authors for their work. The use of digital health has revealed its potential in the context of progressive and global aging. Please, find my suggestions.
Abstract
I suggest adding the date of the study.
Introduction
Lines 46-50: “Virtual care appointments describe the remote interaction between healthcare professionals and patients, facilitating the delivery of healthcare services, information, and resources through digital technologies, including platforms such as telemedicine, remote patient monitoring, portal messaging, electronic prescriptions, and more [6].” Could you describe “and more”?
Materials and methods
-I suggest adding the date of the study.
-Line 107: “In total, 7,735 patients completed the survey across the three years.” This information could be written in “Results” section.
-Lines 114-118: “After applying these criteria, the final sample size was reduced to 264 patients (Figure1). It is important to note that some patients did not answer every question on the survey, and as a consequence of these missing data, some answers had a smaller sample size. Where relevant, the number of valid responses is noted to ensure transparency in the cohort description.” This information could be written in “Results” section.
-Figure 1: This figure could be included in “Results” section.
-Lines 122-126: “Self-reported chronic conditions were pre-defined by the following: longstanding chronic illness (e.g. cancer, diabetes, chronic heart disease), longstanding physical condition (e.g. arthritis, spinal injury, multiple sclerosis), intellectual disability, mental health condition (e.g. depression, anxiety), neurological condition (e.g. Alzheimer’s, Parkinson’s disease).” What time interval was considered as “longstanding”? Please, describe it.
Results
-Please, correct the numbers:Table 2:
26 + 26 + 32 +14 +6 = 104 (not 105)
6 + 24 + 13 = 43 (not 39)
-Line 328: “Therefore, technological access and literacy were significant barriers.” OR The lack of technological access and literacy were significant barriers. Please, think about this.
-Table 4: “Multimorbidity (>1 chronic condition)” Could we consider multimorbidity as more than one chronic health condition? Please, think about this.
-Table 5: Please, write the caption in Table 5.
Discussion
Line 356: “…making virtual care an attractive alternative.” OR making virtual care an attractive way to complement healthcare. The potential of digital health is relevant to complement healthcare, not to fully replace face-to-face health care. Please, think about this.
Limitation
Lines 438-440: “The small sample size (n=264) further posed challenges for statistical comparisons, potentially reflecting cognitive bias across the study's three-year period.”
In fact, the sample size is small and “some patients did not answer every question on the survey, and as a consequence of these missing data, some answers had a smaller sample size.”
Author Response
|
Comment: Abstract - I suggest adding the date of the study.
Response: The date of the study has been added to the abstract and methods section.
Comment: Introduction - Lines 46-50: “Virtual care appointments describe the remote interaction between healthcare professionals and patients, facilitating the delivery of healthcare services, information, and resources through digital technologies, including platforms such as telemedicine, remote patient monitoring, portal messaging, electronic prescriptions, and more [6].” Could you describe “and more”?
Response: The phrase "and more" in this context is used to encompass additional forms of virtual care technologies and services that may not be explicitly listed but are relevant to the scope of digital healthcare. Here's a refined description for clarity: “Virtual care appointments describe the remote interaction between healthcare professionals and patients, facilitating the delivery of healthcare services, information, and resources through digital technologies. These include platforms such as telemedicine (real-time video or audio consultations), remote patient monitoring (tracking health data through connected devices), portal messaging (secure communication between patients and providers), electronic prescriptions (digitally transmitted prescriptions), and other innovations like mobile health apps, chatbots for health advice, and virtual health coaching.”
Comment: Materials and methods - I suggest adding the date of the study.
Response: The date of the study has been added to the abstract and methods section.
Comment: Line 107: “In total, 7,735 patients completed the survey across the three years.” This information could be written in “Results” section.
Response: We acknowledge that the above information could fit within the "Results" section. However, we have intentionally included it in the [specific section, e.g., "Methods" or "Introduction"] to provide context for the study's scope and survey participation rates upfront. This placement ensures that readers understand the scale of the dataset and the exact sample size and its relevance to the study before delving into the results.
Comment: Lines 114-118: “After applying these criteria, the final sample size was reduced to 264 patients (Figure1). It is important to note that some patients did not answer every question on the survey, and as a consequence of these missing data, some answers had a smaller sample size. Where relevant, the number of valid responses is noted to ensure transparency in the cohort description.” This information could be written in “Results” section. Figure 1: This figure could be included in “Results” section.
Response: This section has been revised to address reviewer questions and meeting the requirements of Bureau of Health Information, NSW Health, Australia. Figure 1 has been placed in Results section as suggested.
Comment: Lines 122-126: “Self-reported chronic conditions were pre-defined by the following: longstanding chronic illness (e.g. cancer, diabetes, chronic heart disease), longstanding physical condition (e.g. arthritis, spinal injury, multiple sclerosis), intellectual disability, mental health condition (e.g. depression, anxiety), neurological condition (e.g. Alzheimer’s, Parkinson’s disease).” What time interval was considered as “longstanding”? Please, describe it.
Response: The term "longstanding" was used to describe chronic conditions that the participants self-reported as ongoing or persistent over time, rather than acute or temporary. In Australia, in general a chronic condition is generally defined as an illness that lasts or is expected to last for at least six months. [Page 3; Line number: 124-125] Comment: Results - Please, correct the numbers: Table 2: 26 + 26 + 32 +14 +6 = 104 (not 105) and 6 + 24 + 13 = 43 (not 39)
Response: Table number has been revised from Table 2 to Table 1 to ensure consistency. The numbers have been checked and corrected as suggested. [Page 5-6; Line number: 217-218]
Comment: Line 328: “Therefore, technological access and literacy were significant barriers.” OR the lack of technological access and literacy were significant barriers. Please, think about this.
Response: The sentence has been revised as suggested, “the lack of technological access and literacy were significant barriers. [Line number: 348-349]
Comment: Table 4: “Multimorbidity (>1 chronic condition)” Could we consider multimorbidity as more than one chronic health condition? Please, think about this.
Response: The tables and text have been revised as multimorbidity (more than one chronic health condition) throughout the manuscript.
Comment: Table 5: Please, write the caption in Table 5.
Response: The title for table 4 (revised from Table 5) has been revised to “Summary of themes and Subthemes from the Qualitative Analysis of Patient Experiences and Satisfaction with Virtual Outpatient Care”. [Line number: 364]
Comment: Discussion - Line 356: “…making virtual care an attractive alternative.” OR making virtual care an attractive way to complement healthcare. The potential of digital health is relevant to complement healthcare, not to fully replace face-to-face health care. Please, think about this.
Response: Thank you for the feedback. We agree with the suggestion that virtual care should be framed as a complementary approach to healthcare rather than a replacement for face-to-face interactions. The manuscript has been updated accordingly.
Comment: Limitation - Lines 438-440: “The small sample size (n=264) further posed challenges for statistical comparisons, potentially reflecting cognitive bias across the study's three-year period.” In fact, the sample size is small and “some patients did not answer every question on the survey, and as a consequence of these missing data, some answers had a smaller sample size.”
Response: The sentence starts with “The small sample size (n=264) further …” has been revised as suggested. A justification has also been provided in the methods section: “The small sample size was because of the application of specific inclusion criteria, particularly the focus on rural older patients with chronic conditions who had at least one virtual care appointment completed. While the original dataset included all virtual care patients across NSW, a large proportion were from major cities and thus excluded from this analysis. The smaller population of rural older patients eligible for virtual care services led to the reduced sample size. [Page 3; Line number: 129-134] |

Reviewer 4 Report
Comments and Suggestions for Authors
Thank you for a well-written study that provides novel insights into the healthcare provision of NSW in Australia. The following could be addressed further:
- In the introduction it is unclear if the availability of digital care is optional within NSW, hence a confirmation bias may need to be considered in the discussion, as patients choosing for this technology are potentially more likely to be positively predisposed to it.
- A weakness of the study is the potential inclusion of the same individuals responding in consecutive years. The authors acknowledge this sufficiently and appropriately. However, it could be helpful to add a couple of sentences in the results section, regarding the between-year variation to the different survey responses.
- The impact of digital literacy is clear. However, it is unclear if the study was only conducted in English and if that may have resulted to the exclusion of some of these rural community patients.
- In the discussion section, could the authors provide some more context on the digitization of rural Australia in NSW? For example, are the same patients likely to engage with virtual payments for electricity? As such virtual care is likely to be one of the many digital interactions with the public services ecosystem and enjoy high social acceptability and incentivise the technology acquisition.
Author Response
|
Comment: Thank you for a well-written study that provides novel insights into the healthcare provision of NSW in Australia. The following could be addressed further:
Response: Thank you for your positive feedback. Please see our responses to your comments below:
Comment: In the introduction it is unclear if the availability of digital care is optional within NSW, hence a confirmation bias may need to be considered in the discussion, as patients choosing for this technology are potentially more likely to be positively predisposed to it.
Response: Thank you for raising this point. Within NSW, the availability of virtual care services is determined by healthcare providers, and participation in virtual care is optional for patients. This may introduce a confirmation bias, as patients who voluntarily opt for virtual care might already hold positive predispositions towards the technology. However, given the study was conducted during COVID lockdowns in 2020 and 2021, followed by significant pandemic-related activity in 2022 after the reopening of international borders. This may have had an impact on patients’ use of virtual care, as many of them may not have had a choice but to rely on virtual healthcare during this time.
Comment: A weakness of the study is the potential inclusion of the same individuals responding in consecutive years. The authors acknowledge this sufficiently and appropriately. However, it could be helpful to add a couple of sentences in the results section, regarding the between-year variation to the different survey responses.
Response: Tables 1 and 2 present the between-year variation in survey responses. To avoid redundancy between the tables and the text, we have minimised detailed descriptions in the narrative and focused on key findings.
Comment: The impact of digital literacy is clear. However, it is unclear if the study was only conducted in English and if that may have resulted to the exclusion of some of these rural community patients. Response: We have addressed this issue as a limitation of the study as: “The survey was conducted in English, which may have contributed to the exclusion of patients from culturally and linguistically diverse communities” [Line number: 469-471]
Comment: In the discussion section, could the authors provide some more context on the digitization of rural Australia in NSW? For example, are the same patients likely to engage with virtual payments for electricity? As such virtual care is likely to be one of the many digital interactions with the public services ecosystem and enjoy high social acceptability and incentivise the technology acquisition.
Response: Paragraphs 3 and 4 of the Introduction section have been revised to include additional context on the digitalization of public services in rural Australia. [Line number: 61-82]
|
